# The Task-Based Approach to Teaching Critical Thinking for Computer Science Students

**Elena Mäkiö** *[ORCID] **and Juho Mäkiö**

Department of Electrical Engineering and Informatics, Faculty of Technology,
University of Applied Sciences Emden/Leer, 26723 Emden, Germany
* Correspondence: elena.maekioe@hs-emden-leer.de

**Abstract:** Critical thinking (CT) is one of the most important 21st-century skills that employers believe will grow in prominence. However, many higher education (HE) graduates often lack it. This is also true for graduates in computer science (CS) and related disciplines, who need CT to drive social and organizational digitalization. There are a number of strategies for teaching CT. However, there is no evidence regarding how to effectively teach CT in CS. To address this gap, this study proposes an educational approach that aims to promote CT to the students of CS and related disciplines. An educational experiment using this approach was conducted in two modules with similar content. The written reflections of 11 students on their development in CT and the quantitative data of students' self-assessments of their CT skills and dispositions before (N = 20) and after (N = 11) the experiment were analyzed. Priority was given to the qualitative data. The results of this study support the hypothesis that the proposed approach has a positive impact on the development of students' perceived CT skills. They also show that this approach stimulates and promotes students' ability to transfer CT to other tasks and domains and to other contexts and situations. However, the positive development of students' CT dispositions could not be demonstrated in this study, which can be explained by the short duration of this intervention.

**Keywords:** critical thinking; teaching critical thinking in computer science; educational approach; task-based approach; Socratic questioning

## 1. Introduction

Critical thinking (CT), currently frequently mentioned in society, business, research, and education, is considered one of the 21st-century skills that citizens need in the globalized and digitalized societies [1–3]. Although society and businesses expect higher education (HE) graduates to develop sound disciplinary competence and solid CT skills, many graduates frequently lack CT and the ability to transfer CT skills within and across their subject matter domain [4–7]. Since students' CT skills hardly develop as a side-effect of HE [8], it is the responsibility of HE institutions to promote students' CT in their educational programs and courses through the development and application of research-informed educational concepts and approaches [9]. There is a growing body of literature on how to teach CT skills [7,10–13]. One of the most important findings is that CT is taught efficiently when the general principles of CT and training of CT skills are embedded in a domain-specific course and explicitly stated as course objectives [10].

In our era of societal and organizational digitalization, as well as of Industry 4.0 and 5.0, CT skills are particularly needed by computer science (CS) graduates, who are at the forefront of planning, designing, developing, managing, and applying digital solutions to drive digitalization, digital transformation, and Industry 4.0 and 5.0. It could therefore be expected that the development of CT skills among CS students would be the subject of a number of research studies. The authors of this study reviewed the current literature on how CT is taught in CS and other related disciplines such information systems (IS),

business informatics (BI) and IS management (ISM). The following search strings were applied in the Web of science and Google Scholar digital libraries:

- "critical thinking" + teaching + "computer science",
- "critical thinking" + teaching + "business informatics", and
- "critical thinking" + teaching + "information systems management"

Only publications explicitly dealing with CT teaching in HE were analyzed. There were few of these, which is also confirmed by other scholars [13]. The papers found were exclusively conference papers. A summary of the most important studies is given below.

In [14], the authors aimed to improve CS students' CT by integrating a writing activity in a masters' CS course. In this learning activity, students first researched the concepts presented in this course, summarized their work in writing, and then assessed the work of two fellow students. The authors analyzed the students' written work and concluded that the students improved their CT skills through this activity. In [15], a masters' computer architecture course was redesigned to focus on explicitly teaching CT and research skills. After the general principles of CT were introduced and elaborated through group work, students strengthened their CT skills by writing critical reviews of recently published research. Based on students' self-reports of conceptual understanding, an assessment of the critical reviews, and a final survey of student engagement, the authors concluded that the educational intervention was beneficial to the students.

In the subjects of BI, IS, and IMS, CT teaching was either conducted in separate modules [16,17] or integrated into domain-specific courses, such as 'Systems Analysis and Design' [13,18]. Ref. [18] described the use of subject-related case studies and questioning during discussions to improve students' CT skills. In [13], the authors introduced students to the basic concepts of CT. All authors used student-centered teaching methods, such as class and group discussions of authentic problems and group work. The authors evaluated their teaching interventions through student feedback surveys [17], student self-assessments [15], or by assessing the results of case studies [13] or assignments [14]. They concluded that the interventions were beneficial for the students.

In summary, the studies reviewed describe either the teaching of CT in separate modules or educational interventions introduced that were into domain-specific courses that aim to foster students' CT skills. In their classes, the authors use instructional methods that activate student learning, such as discussions and group work, and specific subject- or discipline-related tasks to educate their students. The authors evaluate the impact of their educational interventions, mainly through student self-assessment surveys, and report positive student satisfaction and the benefits of the learning activities introduced for student learning. The studies reviewed do not address the question of how to promote students' ability to transfer CT, nor do they introduce a general approach to teaching CT, and nor do they use a sound methodology to evaluate their interventions and identify the detailed aspects of changes in students' CT. Therefore, the question of how to teach CT to CS students is still open and is addressed in the present study.

*Research Question*

This study proposes an educational approach in the field of CS that addresses the development of CT and enables the ability to transfer it to other tasks and domains. This study aims to seek responses to the following research question:

RQ: Does the proposed educational approach promote the development of CT, including the ability to transfer CT to the other tasks and domains, in CS students?

This study tests two hypotheses:

1. The proposed approach has a positive impact on the development of students' perceived CT cognitive skills and on their ability to transfer these skills to other tasks and domains.
2. The proposed approach has a positive impact on the development of students' perceived CT dispositions.

## 2. Theoretical Background

### 2.1. Definition of Critical Thinking

CT is frequently mentioned in society, business, research, and education. However, there is no consensus among experts on its definition. For example, Lai [19] distinguishes three approaches to defining CT in education: (1) psychological, (2) philosophical, and (3) educational. Definitions based on the psychological approach relate CT to cognitive skills and understand it as a process. CT is defined as "a metacognitive process that, through purposeful, reflective judgment, increases the chances of producing a logical conclusion to an argument or solution to a problem" ([20], p. 43). Definitions based on the philosophical approach focus on the outcome of CT rather than on the process. CT is defined as "a reflective and reasonable thinking that is focused on deciding what to believe or do" ([21], p. 45). Based on the educational approach, CT is linked to the concept of higher-order thinking from Bloom's taxonomy [22] and located at the higher-order levels of analysis, synthesis, and evaluation.

In this paper, the psychological approach to defining CT is adopted for teaching, and CT is considered as a thinking process. For the purposes of student assessment and the evaluation of interventions, a definition of CT introduced by a Delphi Panel of the American Philosophical Association (APA) is used. CT is "purposeful, self-regulatory judgment which results in interpretation, analysis, evaluation, and inference, as well as explanation of the evidential, conceptual, methodological, criteriological or contextual considerations upon which that judgment is based" ([23], p. 3). This definition presents a framework of six CT cognitive skills: (1) interpretation, (2) analysis, (3) inference, (4) evaluation, (5) explanation, and (6) self-regulation. Additionally, it includes seven CT dispositions: (1) truth-seeking, (2) open-mindedness, (3) analyticity, (4) systematicity, (5) self-confidence, (6) inquisitiveness, and (7) cognitive maturity.

### 2.2. Challenges of Teaching Critical Thinking

Numerous programs and initiatives have focused on teaching CT [9,21]. Many studies emphasize that CT can be promoted through explicit instruction, combined with practice [8,11,12]. However, there are different views on the best teaching strategies and instructional methods to promote the acquisition of CT and the ability to transfer it to other tasks and domains.

Ennis [21] distinguishes four approaches to CT instruction: (1) the general approach, in which CT is taught separately from the presentation of domain-specific content; (2) the infusion approach, which integrates teaching CT in the domain-specific content and makes the CT principles explicit; (3) the immersion approach, which is similar to the infusion but does not make the CT principles explicit; and (4) the mixed approach, which combines the general approach with either the infusion or the immersion approach. In their meta-analysis of studies on CT teaching published between 1960 and 2005, Abrami et al. [10] found that the mixed approach was the most effective strategy for CT teaching, followed by infusion and general approaches. The immersion approach had the smallest effect. A systematic review of CT instruction [12] also indicated a shift towards integrating CT teaching into domain-specific classes.

Two teaching methods—namely, dialogue (or discussion) and engaging students in authentic problems—have been identified as particularly helpful in developing CT [11]. These methods are particularly effective when combined with mentoring (e.g., teacher feedback). Higher-order questioning has been employed as an implicit teaching strategy in some studies (see studies in [12,18]). Some of these studies report that the group of students taught with the higher-order questioning strategy significantly outperformed the group taught with the lower-order questioning strategy in domain-specific CT [12].

Several studies report that the transfer of CT skills to solving similar and dissimilar tasks in the subject area that go beyond a narrow practiced context is an issue [5–7]. Therefore, successful CT teaching means that students will be able to transfer the acquired

CT skills across tasks and domains and to other problems of daily life. In other words, they will be able to apply CT to new contexts and situations [9,19].

Metacognition in the form of conscious reflection on the problem-solving process is important for CT [24,25]. Metacognitive strategies, i.e., behaviors for monitoring and controlling one's thinking processes, can be divided into three categories ([26], p. 254):

1. planning, e.g., "the selection of appropriate strategies, allocation of available resources";
2. monitoring, e.g., "checking task information to validate comprehension, allocating attention to important ideas, and pointing out informational ambiguities";
3. evaluating, e.g., "evaluating one's reasoning, goals and conclusions as well as making revisions when necessary".

The results of Ku & Ho's study [26] show that skilled critical thinkers demonstrate distinct strengths in their ability to plan specific steps that guide thinking (high-level planning) and to revise their approach after evaluation (high-level evaluation). Furthermore, only students with sufficient metacognitive knowledge are able to engage in self-regulatory activities that will lead to improvements in their CT performance.

In engineering and technology education (including CS, IS, BI and ISM), problem-solving and CT are central and crucial components that students need to acquire [27]. CT and problem-solving are the top skills that employers believe will grow in importance by 2025 [3]. The ability to transfer these skills to solving similar and dissimilar problems and tasks within and across domains is particularly important for these graduates. However, the existing literature has shown that CS graduates often lack CT and other transferable skills [28]. Since previous research does not provide a sound and research-informed approach to teaching CT in CS and related disciplines, this study proposes an educational approach that addresses the challenges and issues of state-of-the-art CT teaching and which is based on the research findings on the effective teaching of CT skills. This approach focuses on promoting students' ability to transfer acquired CT skills.

### 2.3. Pedagogical Approaches in HE and the Promotion of CT

In HE, various pedagogical approaches and instructional methods, which are mainly based on constructivist learning principles (see [29]), have been introduced to enable effective learning for students. These approaches support self-directed learning and motivate learners to take personal responsibility for their learning and to collaborate with their peers in the learning process. This active learning also leads to the acquisition of CT skills [30].

Creating meaning to embed new concepts into the existing cognitive structure and forming many meaningful connections of a new concept to other relevant concepts are crucial for the development of CT and the ability to transfer CT to new situations and contexts [25]. The perceptional approach in education [31], which builds on the idea of "meanings first", has influenced the development of the educational approach proposed in this study. The perceptional approach is based on the idea that "perception plays a fundamental role in all learning" ([31], p. 212). Perception is an intuitive, non-conscious process of creating meanings based on empirical observations and interpretations. Perceptional learning evolves from observation and experimentation into comprehension and conceptualization, building knowledge from the lower to higher levels of the conceptual hierarchy.

Different pedagogical approaches aim to develop different knowledge and skills. The acquisition of theoretical knowledge can be facilitated by lectures and presentations, while the development of higher-order cognitive skills can be better triggered by student-centered active approaches such as problem solving [32]. The flipped classroom technique, which is increasingly used in HE, is an approach in which students are expected to read and understand appropriate learning materials prior to a particular teaching session [33]. First, these materials are discussed in class, and then problems or tasks are solved in groups or individually in order to apply and consolidate the acquired knowledge.

The following learning principles are most helpful in efforts to promote the development of CT and the ability to transfer CT skills [24,25]:

1.  Learning general principles and concepts facilitates their transfer to dissimilar problems, as it creates more flexible mental representations;
2.  Abstract generalized principles and the rules of CT need to be linked to varied examples and potential applications in different contexts;
3.  Practices of metacognitive strategies such as self-monitoring, self-awareness, and self-explanations best stimulate learners and promote the transferability of CT skills.

Therefore, not only CT but also metacognitive skills should be promoted in CT teaching. To do this, students need to be encouraged to reflect on their own cognitive activities, e.g., through questioning and feedback from the teacher [25], and metacognitive knowledge has to be explicitly taught when needed [26].

Students elaborate a new idea or concept by making many meaningful connections and relating that concept to other relevant concepts [25]. An effective teaching technique to support this process and to practice recalling a new concept is the use of thoughtful questions [25]. Thoughtful questions, for example in form of Socratic questions, can be also used to engage students in metacognitive activities [34]. "Socratic questioning is systematic, disciplined, and deep and usually focuses on foundational concepts, principles, theories, issues, or problems" ([34], p. 36). This method also implies interest in assessing the truth or plausibility of ideas [34]. In Socratic dialogue, an extension of Socratic questioning, the teacher engages students in discussion and forces them to self-reflect by asking questions [35]. Socratic dialogue cultivates an inner voice of critical reasoning "through an explicit focus on self-directed, disciplined questioning" ([35], p. 36). Facione gathered and categorized higher-order questions according to CT cognitive skills in [30].

Many "authentic" learning tasks and examples need to be used to enhance transfer of CT skills by clarifying and emphasizing the CT skill aspects and their appropriate use [25].

The described pedagogical approaches and teaching and learning principles serve as a background for the proposed educational approach.

### 2.4. The Proposed Educational Approach

The proposed educational approach is based on the evidence on effective teaching and aims to promote both disciplinary and CT skills in CS students and, specifically, their ability to transfer CT skills. Table 1 lists the main features and the corresponding objectives of the approach and refers to the existing evidence.

The proposed approach, which operates at the module level, is based on the principle of perceptional learning [31]. The role of the teacher here is to make learning relevant to the students, to activate their prior knowledge, and to provide an intuitive understanding of what is being taught. Therefore, instruction begins with examples that are relevant to students and help them to first develop an intuitive understanding of the topic in terms of both subject matter and CT concepts. The subsequent introduction of subject-specific concepts and theories and general principles of CT helps students conceptualize new information, place it in their own context, and give it meaning.

The subject matter is divided into several blocks; each block is taught in the following steps:

*   Introduction of subject-specific concepts using the perceptional approach;
*   Processing of a task/problem by students (individually or in group);
*   Discussion of the problem-solving and thinking process and results using Socratic questioning and dialogue;
*   Introduction of general CT principles and aspects or reminder of them if they have already been introduced.

In more detail, the lessons in each block are organized according to the following scheme. First, students are taught subject-specific foundations. Then, students are asked to practice and deepen their theoretical knowledge through problem-solving activities in which students solve tasks individually or in groups. The teacher facilitates a plenary discussion of the solutions by leading Socratic questioning and dialogue with students and asking critical questions (see Table 2) about students' problem-solving and thinking processes. This is to encourage students to self-reflect on their problem-solving activities

and thinking processes. Only after this reflection, students are introduced to the general principles of CT and are able to relate their problem-solving experiences to CT concepts. By the time students complete next tasks, they are aware of the general principles of CT and can consciously apply them. Flipped classroom is used to allow students to engage with the new material prior to class and to use class time for questions and discussions.

**Table 1.** Main features of the proposed approach, the corresponding objectives and reference to the literature.

| Characteristics | Objectives | Reference |
|---|---|---|
| The mixed approach (see [21]) is adopted that integrates the teaching of CT into domain-specific content and makes the CT principles explicit. | To make teaching CT effective within the CS domain. | [10,12] |
| The proposed approach combines the learning of general principles and concepts (both domain-specific and CT) with varied examples in order to anchor these principles in students' own context. | To effectively promote the acquisition of subject-specific skills and the acquisition and transfer of CT. To make learning relevant to students, and to activate their prior knowledge and experience so that they can incorporate the new knowledge into the existing cognitive structure and form a holistic picture. | [24,25,31] |
| Domain-specific concepts and content and general principles of CT are introduced during lectures and presentations. Task/problem-solving activities aim to develop higher-order cognitive skills and CT. The teacher provides feedback on the results. | To effectively promote the acquisition of both domain-specific skills and CT. | [8,11,12,32] |
| Tasks are the central points to promote the development of students' domain-specific skills and CT skills and dispositions. These tasks should include various domain-specific and CT aspects. | To promote the development of domain-specific skills and CT. | [11,25] |
| Socratic questioning and dialogue are used in class to discuss task/problem-solving process, as well as student thinking and outcomes. | To activate students' metacognitive understanding and skills and trigger their reflection on cognitive activities. | [25,26,30,34] |

**Table 2.** The questions of Facione [30] (in bold) and some examples of authors' questions (in italics) used in the experiment and assigned to the CT skills from the Facione's framework.

| Skills | Descriptions | Questions |
|---|---|---|
| Interpretation | "To comprehend and express the meaning or significance of a wide variety of experiences, situations, data, events, judgments, conventions, beliefs, rules, procedures, or criteria" ([23], p. 8). | • **What does this mean?**<br>○ *What does RAMI 4.0 (Reference Architectural Model Industry 4.0) framework mean for the integration of the production chain? (the 'Innovation management' module)*<br>○ *What is the general, overall meaning of this scientific manuscript? (the 'Scientific seminar' module)?*<br>○ *What does "simplicity in technical style" mean? (the 'Scientific seminar' module)*<br>○ *What is understood by "utility" in technical writing? (the 'Scientific seminar' module)*<br>• **What's happening?**<br>○ *What is the impact of the digitalization of society on various professional groups? How can we mitigate the negative impact? (the 'Innovation management' module)*<br>○ *What are the strengths of this manuscript? (the 'Scientific seminar' module)*<br>• **How should we understand that (e.g., what he or she just said)?**<br>○ *What are the consequences of e-business for traditional trade? (the 'Innovation management' module)*<br>• **What is the best way to characterize/categorize/classify this?**<br>○ *What are the common characteristics and what are the differences between EAI (Enterprise Application Integration) and Industry 4.0? (the 'Innovation management' module)*<br>• **In this context, what was intended by saying/doing that?**<br>• **How can we make sense out of this (experience, feeling, or statement)?** |

**Table 2.** *Cont.*

| Skills | Descriptions | Questions |
|---|---|---|
| **Analysis** | "To identify the intended & actual inferential relationships among statements, questions, concepts, descriptions, or other forms of representation intended to express belief, judgment, experiences, reasons, information, or opinions" ([23], p. 9). | • **Tell us again your reasons for making that claim.**<br>　○ *Why should your business idea fulfill the market's needs? (the 'Innovation management' module)*<br>• **What is your conclusion/What is it that you are claiming?**<br>　○ *What are the consequences of the current system architecture for the implementation? Which part of this architecture is assumed to be unchangeable? What are the consequences if these assumptions are not correct? (the 'Innovation management' module)*<br>• **Why do you think that?**<br>　○ *Why do you think that the solution you offer is the solution to the problem you have identified as being the problem? (Often the problem identified is not the one that needs to be solved.) (the 'Innovation management' and 'Scientific seminar' modules)*<br>　○ *Why do you think that "simplicity in technical style" is important in technical documents? (the 'Scientific seminar' module)*<br>• **What are the arguments for and against?**<br>　○ *What are the arguments for and against cooperation with this potential business partner? (the 'Innovation management' module)*<br>　○ *What are the arguments for and against starting production in the Far East? (the 'Innovation management' module)*<br>• **What assumptions must we make to accept that conclusion?**<br>• **What is your basis for saying that?**<br>　○ *What does the author base the conclusion he/she draws in this manuscript on? Could the conclusion have been formulated differently and how? (the 'Innovation management' and 'Scientific seminar' modules)* |
| **Inference** | "To identify and secure elements needed to draw reasonable conclusions; to form conjectures & hypotheses; to consider relevant information & to reduce the consequences flowing from data, statements, principles, evidence, judgments, beliefs, opinions, concepts, descriptions, questions, or other forms of representation" ([23], p. 10). | • **Given what we know so far, what conclusions can we draw?**<br>• **Given what we know so far, what can we rule out?**<br>　○ *After reading the manuscript, which suggestions you would give to the authors to improve it? (the 'Scientific seminar' module)*<br>• **What does this evidence imply?**<br>　○ *After reading and hearing about the health problems associated with the frequent use of mobile technology, what are the expected consequences and risks for people's health based on the information available? (the 'Innovation management' and 'Scientific seminar' modules)*<br>　○ *You have an idea for a research paper. What do you need to justify the importance of the topic for the reader? (the 'Innovation management' and 'Scientific seminar' modules)*<br>• **If we abandoned/accepted that assumption, how would things change?**<br>• **What additional information do we need to resolve this question?**<br>　○ *Let us take a market analysis for a new product: What information are we still missing before we can set up a successful marketing campaign? (the 'Innovation management' and 'Scientific seminar' modules)*<br>• **If we believed these things, what would they imply for us going forward?**<br>• **What are the consequences of doing things that way?**<br>• **What are some alternatives we have not yet explored?**<br>• **Let us consider each option and see where it takes us.**<br>　○ *Let us look at each potential business partner. What are the advantages and disadvantages of working with them? What could be their reasons for working with us?*<br>• **Are there any undesirable consequences that we can and should foresee?** |

**Table 2.** *Cont.*

| Skills | Descriptions | Questions |
|---|---|---|
| Evaluation | "To assess the credibility of statements or other representations that are accounts or descriptions of a person's perception, experience, situation, judgment, belief, or opinion; & to assess the logical strength of the actual or intended inferential relationships among statements, descriptions, questions, or other forms of representation" ([23], p. 9). | • **How credible is this claim?**<br>• **Why do we think we can trust what this person claims?**<br>  ○ *In a research proposal, is the research question coherent and derivable from the text? (the 'Innovation management' and 'Scientific seminar' modules)*<br>• **How strong are those arguments?**<br>• **Do we have our facts right?**<br>  ○ *Is this chain of reasoning plausible and complete? (the 'Innovation management' and 'Scientific seminar' modules)*<br>• **How confident can we be in our conclusion, given what we now know?** |
| Explanation | "To state and to justify that reasoning in terms of the evidential, conceptual, methodological, criteriological & contextual considerations upon which one's results were based; & to present one's reasoning in the form of cogent arguments" ([23], p. 11). | • **What were the specific findings/results of the investigation?**<br>  ○ *You have identified the different customer groups for your proposed product. What are the benefits of the product from the point of view of each customer group?*<br>• **Please tell us how you conducted that analysis.**<br>  ○ *Please tell us how you came up with this business idea, explaininging hy your product should be able to solve the problem you describe?*<br>• **How did you come to that interpretation?**<br>  ○ *You have read the given scientific publication. How did the author arrive at this solution?*<br>  ○ *You have acquired all the information about the disadvantages of this technology. Why is it still widely used in industry and supported by governments?*<br>• **Please take us through your reasoning one more time.**<br>• **Why do you think that (was the right answer/was the solution)?**<br>  ○ *Why do you think the application of Internet of Things technology is beneficial to society?*<br>• **How would you explain why this decision was made?** |
| Self-regulation | Self-consciously to monitor one's cognitive activities, the elements used in those activities, and the results educed, particularly by applying skills in analysis, and evaluation to one's own inferential judgments with a view toward questioning, confirming, validating, or correcting either one's reasoning or one's results ([23], p. 12). | • **Am I precise enough on this?**<br>  ○ *Is the benefit for our potential customers clear enough in your business idea?*<br>• **Is the procedure/methodology good enough?**<br>• **What could I have done better?**<br>  ○ *When I think about my bachelor's thesis, what would I do differently?*<br>  ○ *What should I have done differently to achieve better results in my studies?*<br>• **Is my argument conclusive?**<br>• **How good is my evidence?** |

CT is the way one approaches problems, questions or issues [30] and is an essential tool of inquiry [23]. Problem solving and inquiry are the most important activities that graduates of CS and related disciplines perform in their careers, and they require them to apply their CT skills [27]. Therefore, CS students need to learn effective problem-solving processes and practice them on authentic tasks. For this reason, tasks are important for developing CT and domain-specific skills in CS and are used in the proposed approach. The teacher's role is to monitor student activities, ask critical questions, and provide feedback.



## 3. Methodology

*3.1. Research Design*

To measure the impact of the proposed educational approach on the development of students' CT skills and dispositions and particularly on their ability to transfer CT to other problems and situations, the mixed-methods concurrent triangulation approach [36] was used. Both qualitative and quantitative data were collected concurrently and compared to determine if there were similarities, differences, or a combination thereof. Priority was given to qualitative data.

Within this mixed-method research, the quasi-experimental nonrandomized approach [37] and the one-group pre-test–post-test design [38] were used. An educational experiment was conducted in two modules in which the proposed educational approach was implemented. Since this study was conducted as part of an ERASMUS+ project and the modules were only held in the summer term, it was not possible to form a control group.

Qualitative data were collected at the end of the modules (post-test only) by asking students to answer free-text questions. This was performed to explore how students perceived the development of their CT and subject-specific skills and the change in their understanding of CT. Quantitative data were collected through student self-assessment surveys and in doing so we aimed to explore how students perceived their CT skills and dispositions at the beginning and the end of the modules (pre- and post-test).

It was not the intention of this study to consider its results as representative of the overall population of CS students [39]. Instead, the concept of transferability was used, whereby a comprehensive description of the approach and experiments was provided that could allow the reader to make meaningful relationships to their own situations.

*3.2. Educational Experiment*

To evaluate the proposed educational approach, an educational experiment was conducted in two complete modules. The following modules were designed and developed using the proposed educational approach: (1) the 'Innovation management' module in the master's degree program 'Industrial Informatics—Specialization Industrial Cyber–Physical Systems (ICSP)' and (2) 'Scientific seminar' module in the master's degree program 'Media informatics'. The experiment was conducted in the summer term of 2022 and all contact hours were held online. The experiment ran over the entire term and included 16 interventions of 1.5 h each for each module (one intervention per week). The aim of implementing and conducting these two modules was to collect a greater quantity of data, as only a small number of participants were expected for each module. Other reasons for choosing the two interdisciplinary modules were:

1. The master's degrees of these modules are consecutive degrees to the bachelor's degree in CS at the University of Applied Sciences Emden/Leer (Germany);
2. The intended learning outcomes and topics of these modules offered an opportunity to promote CT in addition to teaching subject-specific skills;
3. The modules included parts with identical intended learning outcomes and topics and were taught using the same teaching materials. Consequently, the collected data were categorized and analyzed together in order to measure the impact of the proposed approach.

Both modules were electives and comprised 5 ECTS points, 21 contact hours and 129 h of self-guided study.

The 'Innovation Management' module consisted of two parts: (1) 'Innovation processes for ICPS', which dealt with the innovation paradigm and methods for its implementation, and (2) 'Creativity techniques and scientific writing'.

The second part of the 'Innovation Management' module and 'Scientific seminar' were identical in terms of their intended learning outcomes and topics, except for some selected tasks and examples that were specific to these degree programs. These teaching units covered topics such as the research process in CS and methods for conducting scientific work, writing academic papers, reading scientific publications, and presenting research

results. Additionally, they promoted methods of creative thinking, selecting ideas, and creative group processes.

### 3.2.1. Teaching in the Experiment

According to the proposed educational approach, the subject matter was introduced to the students using the perceptional approach: first, intuitive understanding and meaning were established. Then, there was a period of discussion and conceptualization. Then, students practiced the acquired knowledge by working on a task. The results, the problem-solving process and the students' way of thinking were discussed using Socratic questioning and dialogue. Table 2 lists the questions that were asked of the students in this experiment. The questions used by Facione [30], which are in bold, were refined by the authors by developing specific questions for these modules (in italics). These refined questions were posed to the students to motivate them to undertake a deeper analysis of specific tasks. Afterwards, the general principles and theoretical aspects of CT (based on the Facione's framework [23]) were introduced to give the students the opportunity to conceptualize the CT principles and place them in the context of the solved task. While working on the following tasks, students were repeatedly reminded to keep the principles and aspects of CT in mind.

The following tasks, which combined subject-specific topics with CT aspects, were given to the students:

1.  Analyze the structure and quality of scientific publications based on the given acceptance criteria for scientific conferences and journals. Apply the quality criteria of conferences and scientific journals to evaluate them. (Students were presented with both published articles and those manuscripts that were submitted but unaccepted in order to demonstrate the difference between high-quality and low-quality texts.)
2.  Critically examine your work on the bachelor's thesis and its results. Answer the following questions: "How did you organize your writing process?" "What would you do differently today and why?" "What have you done well and would not change?" "What would I advise myself to do better?" (The focus of this task was on the CT skill 'self-regulation' and the CT dispositions 'self-confidence' and 'cognitive maturity').
3.  Describe the CT aspects that you used in your bachelor's thesis and the aspects you would use in your bachelor's thesis if you had to do it again. What would you have done differently when working on your bachelor's thesis if you had known the CT principles learned in this module?
4.  Use creative thinking methods in groups to generate innovative ideas in the field of cyber–physical systems (only the 'Innovation Management' module).
5.  Analyze household electricity consumption, investigate how and where energy recovery can be used, identify challenges and describe how to address them. Describe your solution and your personal point of view on the topic, both in writing and orally (both modules).
6.  Analyze benefits and drawbacks of new technologies, e.g., mobile technology, Internet of things technology.

After a class assignment, a similar but more extensive assignment was set as homework. For example, in the 'Innovation management' module, different aspects of the economic and societal consequences of industrial digitalization were briefly discussed using examples. As a homework assignment, the students had to find their own examples and conduct an in-depth analysis in which they had to consider both negative and positive aspects, design elaborate strategies to mitigate negative consequences, and demonstrate the benefits of their use. In the next lesson, each student was asked to briefly describe their solution, and received individual feedback from the teacher. Students participated in the group discussion. Completion of the homework and participation in the class work were included in the grade.

### 3.3. Instruments

A questionnaire with seven questions was used to collect qualitative data (see Table 3). It was developed by the authors based on their experience as teachers and is validated in this research. As the questions of the questionnaire address general aspects of CT (based on the Facione's framework [23]) and are subject-neutral, they can be used across subjects in CS. Students gave their free text responses, reflecting on their learning experiences and outcomes.

**Table 3.** The questionnaire to collect qualitative data.

| Scale | Questions |
| --- | --- |
| CT: definition and process | 1. What does it mean to you to 'think critically' (when trying to solve a problem or deal with information)?<br>2. Describe what you do to solve a problem. Explain the steps. |
| Professional and personal development | 3. What have you learned for your professional and personal development by participating in this module? |
| Understanding of CT and CT skills | How have your understanding and skills changed due to this module? Consider the following aspects:<br>4. Understanding of the subject;<br>5. Understanding of critical thinking;<br>6. The link between theory and practice;<br>7. The ability to solve problems. |

Two questionnaires were used to collect quantitative data. They included the scales that reflect Facione's framework [23] of CT skills and dispositions. A 60-item questionnaire (a short version of Critical Thinking Self-Assessment Scale—**CTSAS**) [40] assessed students' perceptions of CT skills, and a 21-item questionnaire (Student-Educator Negotiated Critical Thinking Dispositions Scale—**SENCTDS**) [41] assessed students' perceptions of CT dispositions. The internal consistency of the **CTSAS** questionnaire is excellent (Cronbach's $\alpha = 0.969$) [40], and that of the **SENCTDS** questionnaire is good (Cronbach's $\alpha = 0.773$) [41]. Both questionnaires were arranged on a 7-point Likert-type scale with the following values:

- 'Never' = 1; 'Rarely' = 2; 'Occasionally' = 3; 'Usually' = 4; 'Often' = 5; 'Frequently' = 6; 'Always' = 7;
- 'Strongly Disagree' = 1; 'Disagree' = 2; 'Slightly Disagree' = 3; 'Neither Agree nor Disagree' = 4; 'Slightly Agree' = 5; 'Agree' = 6; 'Strongly Agree' = 7

### 3.4. Data Collection and Analysis

In two modules with similar content (see Section 3.2), an experiment was conducted and data were collected. The students who participated in the modules were asked to answer an online questionnaire during the first and the last lecture and to write a reflection as part of final exam by answering the questions in Table 3. The students who participated in the data collection signed an informed consent form about their participation in this study. Table 4 lists the number of students' responses collected. Only 42% and 55% of students who attended the first lecture in the 'Innovation management' module and 'Scientific seminar', respectively, completed the corresponding module. The reasons for this were that many students changed their minds about attending the modules or postponed their attendance until the next academic year because the modules were electives and offered online (due to COVID-19 limitations).

**Table 4.** Sample sizes in quantitative and qualitative research.

| Module | Number of Participants at the Beginning | Sample Size—Quantitative Survey | | Number of Paired Responses | Number of Written Reflections (Qualitative Research) |
|---|---|---|---|---|---|
| | | Pre-Test | Post-Test | Pre/Post | Post |
| Innovation Management | 12 | 10 | 5 | 3 | 5 |
| Scientific seminar | 11 | 10 | 6 | 5 | 6 |

The sample size of the quantitative survey was too small for any statistical conclusions to be drawn. Given the precedents in similar studies [15,42,43] and the richness of the qualitative data, a sample of 11 students is reasonable for transferring the results of the analysis to other modules with a similar educational approach.

The students' written reflections were transcribed, coded and analyzed using the General Inductive Approach (GIA) [44] on NVivo analysis software to uncover and describe the most important categories and themes. During coding, the qualitative data were reduced and analytically categorized. Based on this categorization, qualitative explanations and generalizations were developed that remained close to the concrete data and contexts but went beyond simple summary descriptions of the data [45]. Therefore, the qualitative findings can be transferred to other, similar situations and to other, similar groups of students.

The responses to the quantitative pre-test and post-test surveys were paired, resulting in eight pairs. Due to the small data sample (less than 10), single-subject research and its methods of data analysis were used [46]: (1) chats were created to visually inspect the data, and (2) descriptive statistics were calculated for all scales of the **CTSAS** and **SENCTDS** questionnaires using IBM SPPS statistical software. Hedges' g (for sample sizes < 20) was estimated to measure effect size as the practical significance and impact of the educational intervention on student learning. Hattie [47] provides a useful benchmark for evaluating such impacts. The average effect size of educational interventions is 0.4. Effect sizes of less than 0.2 indicate a lack of teaching, effects between 0.2 and 0.6 mean medium teaching effects, and those greater than 0.6 are considered large when judging educational outcomes.

## 4. Results

### 4.1. Students' Reflections on the Development of Skills

Four broad categories were identified from the GIA of the reflections, mainly due to the questions of the qualitative questionnaire (s. Table 3): (C1) What is CT for me?; (C2) What do I do to solve a problem?; (C3) What did I learn in the module?; and (C4) How did my understanding and skills change?. The most prominent themes (mentioned by more than 30% of students) were grouped under these categories, as shown in Figure 1. All other themes appeared in less than 30% of the reflections, and so only the most interesting of them are mentioned separately.

The students' reflections varied in length, detail, and quality. Table 5 shows the number of words and codes per category for each student reflection. The reflections of students S1 and S2 are not written in the first person and are descriptive rather than analytical. This is due to the fact that students in technical subjects often have difficulty when writing reflections (e.g., see [48]).

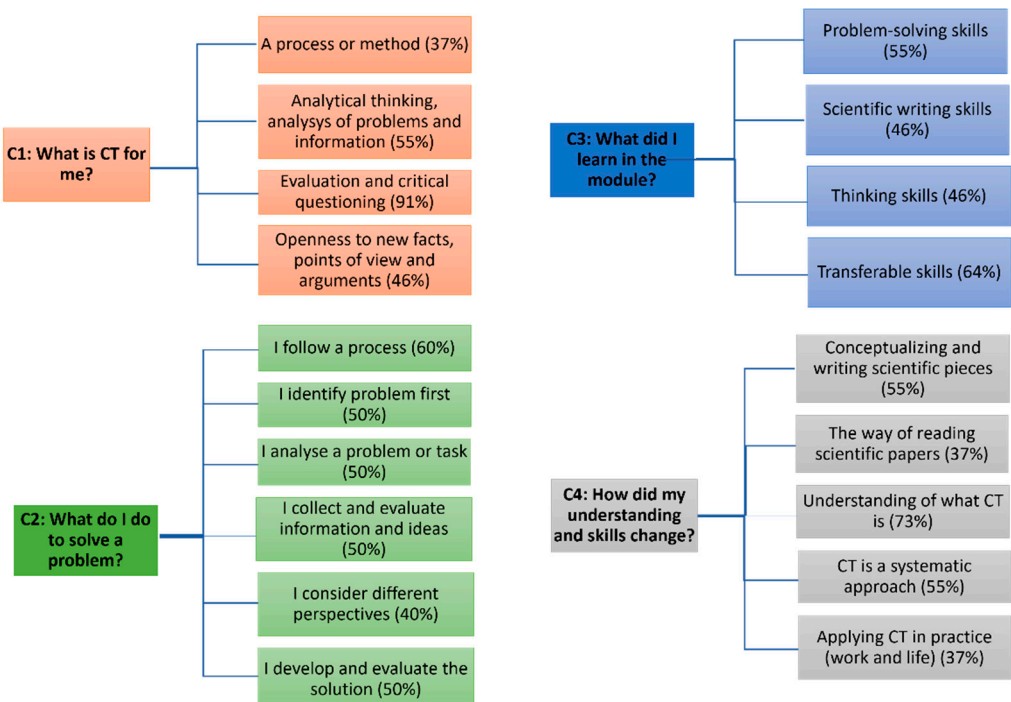

**Figure 1.** Analysis of student reflections (N = 11), broken into 4 categories (C1–C4), with the 19 most common themes shown nested under these categories. The percentage of students who named the specific theme is given in brackets.

**Table 5.** Number of words, codes created and codes per category for each student reflection.

| Reflection of Student # | Number of Words | # Codes in C1 | # Codes in C2 | # Codes in C3 | # Codes in C4 | Comment |
|---|---|---|---|---|---|---|
| Innovation Management | | | | | | |
| S1 | 1033 | 8 | 2 | 6 | 7 | Some formulations of this reflection did not feel like student's own experience |
| S2 | 530 | 3 | 1 | 1 | 3 | Some formulations of this reflection did not feel like student's own experience |
| S3 | 1448 | 10 | 9 | 6 | 9 | |
| S4 | 613 | 3 | 1 | 1 | 2 | This student has a controversial understanding of CT |
| S5 | 2338 | 2 | 13 | 11 | 14 | |
| Scientific seminar | | | | | | |
| S6 | 690 | 2 | 9 | 9 | 3 | |
| S7 | 1184 | 6 | 12 | 7 | 5 | |
| S8 | 512 | 7 | 0 | 5 | 1 | This student did not answer question 2 [1] |
| S9 | 1148 | 8 | 7 | 8 | 13 | |
| S10 | 649 | 4 | 7 | 2 | 1 | |
| S11 | 564 | 4 | 8 | 4 | 3 | |

[1] For question 2 see Table 3.

### 4.1.1. C1: What Is CT for Me?

In category C1, the majority of students (91%) defined CT as the ability to evaluate and critically question existing theories and information ('*critically questioning one's own assumptions as well as those of others*'—S11). A total of 55% of the students named the CT skill as the ability to think analytically and analyze problems and information. Overall, 55% of the students perceived CT as a specific personal disposition. Some 46% of the students named openness to new facts, points of view and arguments ('*it's about being open-minded*'—S3) as the ability to think critically, while 19% named skepticism and curiosity

as characteristic of CT. In total, 37% of the students explicitly described CT as a process or methodology ('*critical thinking is a process by which I can approach and deal with a problem or topic*'—S10) and as the ability to make reasoned judgments about problems or tasks ('*Making logical, well-considered decisions out of reasoned judgments*'—S1). Some students included in CT the ability to understand counterarguments of others and to be impartial and objective.

### 4.1.2. C2: What Do I Do to Solve a Problem?

In category C2, the majority of students (60%) explicitly stated that they follow a strict process when solving problems. In this process, 50% of the students noted that they first identify a problem ('*First, the problem must be identified*'—S7); 50% of the students indicated that they analyze the problem; 40% of the students indicated that they break down the problem into sub-problems ('*divide the problems into sub-problems to make them easier to deal with*'—S6). Some 50% of the students collect the existing information and generate several ideas to solve the problem. When finding a solution, students develop, evaluate and validate the solution (50%) by considering different perspectives (40%) ('*You have to ask yourself whether the problem has been looked at from all possible perspectives*'—S11).

### 4.1.3. C3: What Did I Learn in the Module?

In category C3, the students reported developing various skills, which were grouped as follows (the percentage of students who developed a particular group of skills is shown in brackets):

1. Subject-specific skills (73%);
2. Transferable skills (64%);
3. Problem-solving skills (55%);
4. Thinking skills (46%);
5. Metacognitive skills (28%).

In terms of subject-specific skills, 55% of the students believe that they developed the skill of academic writing. The students listed many different transferable skills they acquired, for instance, creativity and curiosity, openness to different ideas, opinions and methods, and a better understanding of other perspectives. Some 37% of the students believe that they learned to solve problems better and more systematically ('*I definitely learned to work on problems more systematically*'—S6), while 28% of the students believe that they acquired analytical skills ('*Now I can easily relate to the problems and their solutions by using analysis skills that I gained from this module*'—S3) and learned a systematic approach to thinking ('*I am able to analyze these ideas and adjust them systematically*'—S1). Some students believe that they improved their metacognitive skills ('*the consistent approach of critical thinking is also a learning process that I will continue to consolidate*'—S11).

### 4.1.4. C4: How Did My Understanding and Skills Change?

In category C4, students reflected on the change in their understanding and skills in relation to different aspects. The majority of students (73%) reported that their understanding of what CT is has changed significantly and irreversibly: '*My understanding of critical thinking has changed to such an extent that I no longer associate the term with just questioning statements*' (S10); '*now I think Critical thinking outlays one's ability to think differently and connect the dots of the problem*' (S3); '*my understanding has changed fundamentally here*' (S5). Some 55% of the students reported that they recognized CT as a systematic process or approach to solving problems ('*Before, critical thinking for me was just questioning information without a systematic approach*'—S6). In total, 55% of the students indicated that their understanding and skills in conceptualizing and writing academic and scientific pieces have improved ('*I improved my skillset on how to write a paper*'—S4, '*My understanding of creative writing has changed*'—S3). Overall, 37% of the students reported that their way of reading scientific papers has changed ('*Fast reading of papers is possible and legitimate. Before, I always assumed that an 8-page paper in English always takes 8 h to read*'—S9). Indeed, 37% of the students

have applied the CT skills acquired in the module or see the benefit of such application in their job, private projects or hobbies, i.e., they can transfer CT ('*When conducting another project work, I successively tried to consider the aspects of critical thinking from the beginning and to approach a problem solution in a reflective manner*'—S11; '*In relation to the sudden occurrence of critical and demanding problems, I now feel well prepared.*'—S5; '*In the personal area. . . from this module will help me in particular: critical thinking and the writing process, especially when creating and sketching electrical circuits that I build myself*'—S5; '*Especially in agile software development, in which I work professionally, it <CT> will help me to find better solutions faster in a team*'—S9). Some students mentioned that their creativity and understanding of creativity and motivation changed. Overall, 28% of the students reflected on the new understanding of their strengths and weaknesses, as well as on the development of their CT ('*help me avoid any kind of negative or limiting beliefs, and focus more on my strengths*'—S1).

### 4.2. Students' Self-Assessment of Critical Thinking Skills and Dispositions

The student self-assessment surveys yielded 8 pairs of responses in two modules. An analysis of these pairs revealed some changes in students' CT skills and dispositions. Figure 2 shows the individual scores of all respondents in the pre- and post-test measurements. Visual inspection of the charts shows that most students assessed almost all of their CT skills and dispositions at the same high level, except for two dispositions: attentiveness and open-mindedness, which were much lower. Most students' scores improved after the experiment, and some improved significantly. Two students significantly worsened their assessment of attentiveness and open-mindedness. One student assessed almost all CT dispositions significantly as being worse after the experiment, which had a negative effect on the mean scores of the scales.

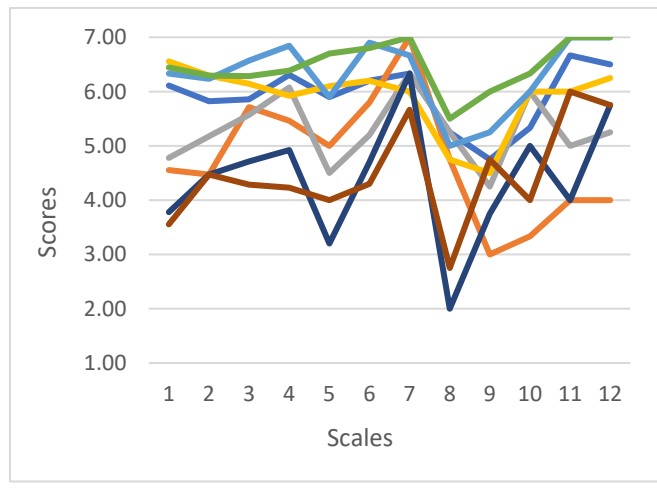

(**a**) Pre-test measurement

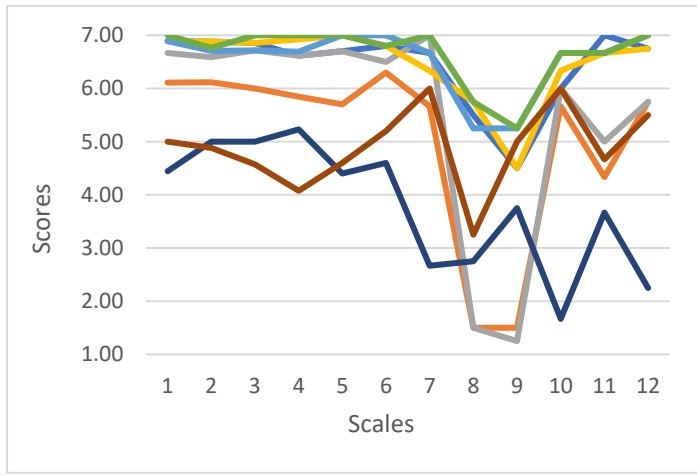

(**b**) Post-test measurement

**Figure 2.** Individual participants' scores on the scales. The vertical axis—scores on a 7-point Likert-type scale (see Section 3.3). The horizontal axis—**CTSAS** scales: 1 = interpretation, 2 = analysis, 3 = evaluation, 4 = inference, 5 = explanation, 6 = self-regulation; **SENCTDS** scales: 7 = reflection, 8 = attentiveness, 9 = open-mindedness, 10 = organization, 11 = perseverance, 12 = intrinsic goal motivation.

As no meaningful differences were found between the scores of the **CTSAS** and **SENCTDS** scales in the two modules, the mean and standard deviation, mean difference and effect size (Hedge's g) were calculated for each questionnaire scale for the pre- and post-tests for both modules together (s. Table 6). The scores for the scales were calculated based on the average of the student responses to the questions assigned to each scale.

**Table 6.** Mean, standard deviation, mean difference and effect size for paired responses of the pre- and post-tests on the **CTSAS** and **SENCTDS** scales. The names of the **CTSAS** and **SENCTDS** scales are in bold for readability purposes. Negative mean differences are highlighted with the background color.

| Scales | No of Pairs | Pre-Test | | Post-Test | | Mean Difference | Effect Size |
|---|---|---|---|---|---|---|---|
| | | Mean | Std.Dev. | Mean | Std.Dev. | | Hedges' g |
| **CTSAS (skills)** | 8 | | | | | | |
| Interpretation | | 5.26 | 1.24 | 6.24 | 0.99 | 0.97 | 0.873 |
| Analysis | | 5.40 | 0.86 | 6.23 | 0.83 | 0.82 | 0.982 |
| Evaluation | | 5.64 | 0.78 | 6.21 | 0.94 | 0.57 | 0.660 |
| Inference | | 5.77 | 0.86 | 6.13 | 1.02 | 0.36 | 0.382 |
| Explanation | | 5.16 | 1.20 | 6.14 | 1.10 | 0.98 | 0.851 |
| Self-regulation | | 5.76 | 0.95 | 6.25 | 0.88 | 0.49 | 0.535 |
| **SENCTDS (dispositions)** | 8 | | | | | | |
| Reflection | | 6.42 | 0.46 | 6.00 | 1.43 | −0.42 | 0.395 |
| Attentiveness | | 4.41 | 1.30 | 3.91 | 1.87 | −0.50 | 0.310 |
| Open-mindedness | | 4.53 | 0.91 | 3.88 | 1.62 | −0.66 | 0.495 |
| Organization | | 5.25 | 1.08 | 5.63 | 1.64 | 0.38 | 0.274 |
| Perseverance | | 5.71 | 1.24 | 5.58 | 1.31 | −0.13 | 0.102 |
| Intrinsic goal motivation | | 5.94 | 1.00 | 5.84 | 1.58 | −0.09 | 0.076 |

The average scores for the CT skills scales were higher in the post-test than in the pre-test sets, while most of the average scores for the CT dispositions skills were lower. Responses on all scales for the CT dispositions were significantly more dispersed in the post-test than in the pre-test era. CT skills improved with a large and medium treatment effect due to the use of the proposed educational approach, while some CT dispositions decreased with a medium treatment effect.

Five students in the sample participated in both the quantitative pre- and post-tests and the qualitative reflections. Tables 7 and 8 show the CT skills and disposition scores of these students correspondingly and highlight the scores that remained the same or worsened. It is worth noting that there is a clear tendency that for most of the CT skills scales (Table 7), students' scores were higher in the post-test than in the pre-test period. In contrast, for the CT disposition scales (Table 8), there is no tendency observed in student scores, and they are more mixed in the pre- and post-tests.

**Table 7.** CT skills scores (**CTSAS**) for the pre- and post- tests of the students who participated in the qualitative research.

| Students | Interpretation | | Analysis | | Evaluation | | Inference | | Explanation | | Self-Regulation | |
|---|---|---|---|---|---|---|---|---|---|---|---|---|
| | Pre | Post | Pre | Post | Pre | Post | Pre | Post | Pre | Post | Pre | Post |
| S1 [1] | 6.44 | 7.00 | 6.29 | 6.76 | 6.29 | 7.00 | 6.38 | 7.00 | 6.70 | 7.00 | 6.80 | 6.80 |
| S2 [1] | 4.56 | 6.11 | 4.47 | 6.12 | 5.71 | 6.00 | 5.46 | 5.85 | 5.00 | 5.70 | 5.80 | 6.30 |
| S5 [1] | 6.11 | 6.89 | 5.82 | 6.88 | 5.68 | 6.86 | 6.31 | 6.62 | 5.90 | 6.70 | 6.20 | 6.80 |
| S7 [2] | 3.78 | 4.44 | 4.47 | 5.00 | 4.71 | 5.00 | 4.92 | 5.23 | 3.20 | 4.40 | 4.70 | 4.60 |
| S11 [2] | 3.56 | 5.00 | 4.47 | 4.88 | 4.29 | 4.57 | 4.23 | 4.08 | 4.00 | 4.60 | 4.30 | 5.20 |

[1]—Innovation management. [2]—Scientific seminar.

**Table 8.** CT dispositions scores (**SENCTDS**) for the pre- and post- tests of the students who participated in the qualitative research.

| Students | Reflection | | Attentiveness | | Open-Mindedness | | Organization | | Perseverance | | Intrinsic Motivation | |
|---|---|---|---|---|---|---|---|---|---|---|---|---|
| | Pre | Post | Pre | Post | Pre | Post | Pre | Post | Pre | Post | Pre | Post |
| S1 [1] | 7.00 | 7.00 | 5.50 | 5.75 | 6.00 | 5.25 | 6.33 | 6.67 | 7.00 | 6.67 | 7.00 | 7.00 |
| S2 [1] | 7.00 | 5.67 | 4.75 | 1.50 | 3.00 | 1.50 | 3.33 | 5.67 | 4.00 | 4.33 | 4.00 | 5.75 |
| S5 [1] | 6.33 | 6.67 | 5.25 | 5.50 | 4.75 | 4.50 | 5.33 | 6.00 | 6.67 | 7.00 | 6.50 | 6.75 |
| S7 [2] | 6.33 | 2.67 | 2.00 | 2.75 | 3.75 | 3.75 | 5.00 | 1.67 | 4.00 | 3.67 | 5.75 | 2.25 |
| S11 [2] | 5.67 | 6.00 | 2.75 | 3.25 | 4.75 | 5.00 | 4.00 | 6.00 | 6.00 | 4.67 | 5.75 | 5.50 |

[1]—Innovation management. [2]—Scientific seminar.

## 5. Discussion

Although various programs and courses have been introduced in recent decades for the purpose of teaching CT, many HE graduates lack this skill. In CS and related disciplines in which the ability to think critically is one of the central components of professional activity, the teaching of CT is rarely and insufficiently addressed. The objective of this study was to introduce an educational approach to teaching CT in CS and related disciplines and to explore its effectiveness. This approach aims to promote subject-specific and CT skills and the ability to transfer CT. This study describes an educational experiment in which the approach was used, and also collects and analyses qualitative and quantitative data on students' perceptions.

Studies of CT teaching in CS and other related disciplines [13–18] have shown that addressing CT topics in discipline-specific modules, both by explicitly introducing general CT principles and by reinforcing CT using discipline-specific cases, is beneficial for developing students' CT skills. While these studies use different teaching approaches and learning activities to promote CT, they have a commonality of using active learning strategies. The results of this study confirm the findings of previous research, i.e., it uses a research-informed activating educational approach and demonstrates through a mixed-methods research design that students have improved their CT skills. As the previous studies did not use a standard measure to evaluate the approaches used, it is not possible to compare their results with those of the present study.

Considering the qualitative and quantitative data obtained and analyzed in this study, the following three points are raised for discussion: (1) students' understanding of CT and the problem-solving process, (2) what students learned in the experiment, and (3) the change in students' understanding and skills as a result of this experiment.

### 5.1. Students' Understanding of CT and the Problem-Solving Process

The students participating in the experiment perceived CT as a cognitive skill in their reflections. Most of them defined CT as the ability to evaluate and question existing information, arguments, claims and theories. They associated CT with analytical thinking that taking apart problems and existing information in order to better analyze and understand them. Some students defined CT as finding the optimal or best possible solution to a problem. These definitions and the aspects of CT mentioned by students are consistent with the definitions of CT as a set of cognitive skills given by Facione [23]. Many students believe that CT has another component—CT dispositions—and named openness to new facts, different points of view, and original perspectives as important aspects of CT aspect. This is also in line with Facione's definition. The openness skill, coupled with cognitive CT skills, forms a stable background for a successful career in CS and related disciplines (see [49]).

While some students explicitly define CT as a process or method, most students believe that one must follow a process to effectively solve a problem or task. They describe this process as consisting of several steps that correspond to Facione's 5-Step CT general problem-solving process [30] and the CT process described by Nosich [50].

Therefore, in terms of students' perception of CT and their approach to task/problem solving, the proposed educational approach was effective and had a positive impact on students' understanding of CT.

## 5.2. What Students Learned

Most students described the subject-specific skills they acquired during the experiment. They mentioned (1) writing skills, including methods for structuring scientific papers and academic writing, and (2) methods for anticipating and writing a master's thesis, with a focus on defining a research question. This was the intended learning outcome of the modules; therefore, the proposed approach successfully addressed the promotion of students' subject-specific skills.

The students reflected extensively on the development of their thinking skills, methods and techniques. They reported that they learned to approach problem solving better and more systematically, acquired the ability to identify and structure problems and to consciously follow the problem-solving process. Some students indicated that they acquired analytical skills, learned a systematic approach to thinking, and were able to sort their thoughts better. The findings obtained from the student reflections were confirmed by the results of the quantitative surveys on the students' perceptions of CT skills (see Tables 6 and 7). In fact, the average scores of the students' CT skills were higher after the experiment than before (see Table 6). These results confirm the findings of similar educational experiments conducted in other subjects [51,52] and are consistent with the results reported in the experiments in CS, even though these results were obtained with other, non-validated instruments, e.g., [13–15,17].

Many students emphasized that they developed their transferable and metacognitive skills (see Section 4.1.3). They became more open to other ideas, opinions and methods, and were better able to understand other perspectives. They learned to be more creative and developed an understanding that curiosity is effective and useful for both problem solving and CT. They learned to present their thoughts in front of an audience and to find the best solution through discussion. Furthermore, students emphasized that, due to the experiment, they adopted the practice of reflecting on their own thoughts when solving problems. Some students compared their thinking habits before and after the experiment: some of them concluded that their habits improved significantly and that they became more systematic in their thinking, while others were convinced of the correctness of their original thinking and problem-solving methods. One student specifically pointed out that CT needs to be practiced continuously and consciously in order to become a good critical thinker. This indicates that the students experienced positive and lasting changes in their thinking, transferable skills, and metacognitive development. This statement is also supported by the results of the **CTSAS** survey, which showed a significant improvement in students' perceived CT cognitive skills (Tables 6 and 7). However, these qualitative findings are contradictory to the results of the quantitative **SENCTDS** survey on students' perceptions of CT dispositions (Table 8).

The results of the quantitative **SENCTDS** survey on students' CT disposition showed some ambiguity: some of them remained stable or slightly decreased at the end of the experiment, others became significantly lower compared to the values observed at the beginning of the experiment. These results confirm the ambiguity of the results obtained in similar studies. While some studies report only few statistically significant changes in students' CT dispositions, including their decrease [53], other studies report an increase in CT dispositions [54] or even their significant improvement [55]. However, as different studies use different instruments to measure CT dispositions (e.g., the current study, [53–55]), the comparison of results may not be absolutely valid. Interestingly, the results on the "open-mindedness" scale confirm the earlier finding that CS students are significantly more ambivalent and less positive on this scale than humanities students [53]. The results obtained in this study can be explained by the fact that the students had a relatively high baseline level of dispositions (>3.5 in 7-point Likert scale) (see [53]) and also by factors

such as the development of students' abilities in CT during the experiment and the short duration of the experiment. These are significant as changing dispositions requires more time and effort, as well as repeated practice [25,53].

Based on the qualitative and quantitative results, we can therefore conclude that the proposed approach implemented in this experiment had a positive impact on the development of students' CT cognitive skills, but not necessarily on their CT dispositions.

### 5.3. Change in Students' Understanding and Skills

The authors have not found any literature, especially in the field of CS and related disciplines, that has reported and analyzed students' perceptions of changes in their understanding and skills in CT or in their ability to transfer CT to other fields. Therefore, the current study seems to present unique findings on this topic in the area of CS education.

The experiment conducted led to changes in students' understanding in the fields of the subject domain, CT, and personal development. In the subject domain, many students reported that they changed the way they read and analyze scientific papers and concluded that reading others' scientific papers enriched their own writing skills. Their understanding of writing their own academic texts changed as they became aware of methods for conceptualizing and writing academic pieces, such as considering the target audience. Students emphasized that they gained an understanding of scientific work and research in CS. Some students reported that they were able to transfer this understanding to their own projects and work. All this information speaks to the students' acquired ability to transfer their subject-specific knowledge and skills to other contexts and fields.

In the area of transferable skills, some students reported a change in their behavior by becoming more skeptical, creative and curious. Some students reflected on their strengths and weaknesses with regard to CT and their development process as critical thinkers during the experiment.

It is worth noting that in their reflections, students more frequently reported a change in their understanding of CT and their development towards becoming critical thinkers. For most students, the definition and meaning of CT was a new insight that made them realize that CT is a combination of skills and dispositions, as well as a method of thought that can be learned and practiced. They associated CT with a systematic approach and problem-solving process. Most students reported a change in their approach to analyzing and evaluating a problem or data, emphasizing that they began to consider unfavorable solutions and base their thoughts only on facts. Based on these findings, it can be concluded that the proposed educational approach applied in the experiment resulted in a change of students' cognitive CT skills, which was confirmed by the quantitative results (see Section 4.2).

However, the most important finding is that 37% of the students explicitly reported that they transferred the acquired CT strategies, methods and ways of thinking to other domains and problems that were extra-curricular and not covered by the modules. Specifically, they used their skills to: (1) lead agile software development teams and involve other team members in the CT process to facilitate their work; (2) solve problems in the field of electrical engineering; (3) conceptualize and write their master's thesis; and (4) to ask questions in everyday life, e.g., in the field of space and climate change. The students' ability to transfer CT to other contexts was enhanced by solving various tasks and assignments given to the students during the experiment (see Section 3.2.1). All this demonstrates that the proposed educational approach applied in this experiment led to the development of students' ability to transfer CT to other domains, problems and contexts and to their understanding of the importance of CT for their work, own projects, or hobbies.

### 5.4. Study Limitations

A limitation of this study is the small number of students who participated in the qualitative research (11 student reflections) and the small number of paired survey responses from these students (5 pairs). Another limitation is that the proposed approach has only been implemented in two modules so far.

*5.5. Future Works*

In order to obtain more evidence for the proposed approach, future studies are planned. Specifically, we will: (1) repeat the educational experiment in the modules described in this study in order to collect more data; (2) implement the "Programming" module in the bachelor's degree in CS using the proposed educational approach; and (3) compare the proposed approach with an alternative approach in the "Programming" module (control group). In addition, the qualitative questionnaire developed by the authors for this study will be validated.

## 6. Conclusions

This research study proposed an educational approach for use in CS education that aims to promote CT and the ability to transfer CT. An educational experiment using this approach was conducted in two modules with similar content. The written reflections of 11 students on their CT development and the quantitative data of students' self-assessments of their CT skills and dispositions were analyzed.

Based on the qualitative and quantitative results, we conclude that the proposed approach had a positive impact on the development of students' CT cognitive skills, which supports the first hypothesis. The results also demonstrate that this approach stimulates and promotes the ability to transfer CT to other tasks and domains and to other contexts and situations. However, the second hypothesis on the development of CT dispositions could not be fully supported by this educational experiment. The positive development of the students' CT dispositions could not be demonstrated in this study, which can be explained by the short duration of this intervention.

**Author Contributions:** Conceptualization, E.M. and J.M.; methodology and formal analysis, E.M. and J.M.; investigation, E.M.; resources and data curation, E.M.; writing—original draft preparation, E.M.; writing, E.M.; review and editing, J.M.; visualization, E.M. All authors have read and agreed to the published version of the manuscript.

**Funding:** This work has been supported by the "Critical Thinking for Successful Jobs—Think4Jobs" Project, with the reference number 2020-1-EL01-KA203078797, funded by the European Commission/EACEA, through the ERASMUS + Programme. The European Commission support for the production of this publication does not constitute an endorsement of the contents, which reflect the views only of the authors, and the Commission cannot be held responsible for any use which may be made of the information contained therein.

**Institutional Review Board Statement:** The study was conducted in accordance with the Declaration of Helsinki, and approved by the Institutional Review Board of University of Applied Sciences Emden/Leer.

**Informed Consent Statement:** Informed consent was obtained from all subjects involved in the study.

**Data Availability Statement:** Not applicable.

**Conflicts of Interest:** The authors declare no conflict of interest.

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
