# Peer review of "The Task-Based Approach to Teaching Critical Thinking for Computer Science Students"

_education, doi:10.3390/educsci13070742_

Round 1
Reviewer 1 Report (New Reviewer)
Lines 194-198 - Authors should provide references.
Line 199 - Distinguish near transfer versus far transfer.
Line 532- Table 6 - Why are the mean differences and effect sizes not shown?
- Lines 551... - Discussion - Authors should present studies that confirm or not their results.
- Line 670 - Study limitations - Possible Hawthorne effect?
Author Response
Dear Reviewer,
thank you for your comments. Please see the attachment.
Best regards

Reviewer 2 Report (New Reviewer)
The article addresses a problem of interest in higher education. The review of the literature and the theoretical references are adequate to support the formative proposal that they describe.
Given the low number of participants, I recommend presenting the study as a pilot in which the training proposal is implemented. In this sense, the data could be critically analyzed to suggest improvements for a new implementation. For example, are modifications required to the instrument that collects qualitative data?
The questionnaire to collect qualitative data is not described in detail, the following doubts arise: 1) It was created by the authors, but how was the validation process carried out? 2) In the first item of the first scale, what is the question or instruction? What defines critical thinking and describes its process?
In the discussion section of the results, especially sections 5.2 and 5.3; there is effectively no relationship between the data presented and previous studies.
In the conclusions section, it is recommended to make explicit the contributions of the study presented. What gaps found in the literature are answered in this research?
The methodology does not mention whether the participants signed an informed consent about their participation in this study.
Author Response
Dear Reviewer,
thank you for your comments. Please see the attachment.
Best regards

Reviewer 3 Report (New Reviewer)

Author Response
Dear Reviewer!
Thank you for your valuable comments! Please see the attachment.
Best regards

Round 2
Reviewer 3 Report (New Reviewer)
The authors adequately addressed the comments of the first review report.
This manuscript is a resubmission of an earlier submission. The following is a list of the peer review reports and author responses from that submission.
Round 1
Reviewer 1 Report
1. This study should be addressed as a single-subject study because of the small sample size.
2. Theoretical framework does not support the background of the study.
3. There is a lack of pedagogical supporting ideas on how the authors designed teaching methods that would promote critical thinking.
4. Discussion could be improved by connecting results of the study and the findings from the previous studies.
Author Response
Dear Reviewer,
thank you for your comments.
We have uploaded our response as a file

Reviewer 2 Report
This work addresses the important issue of teaching critical thinking for computer science students. The purpose of this study is transcribed in the research question (lines 86-87)
Line 84 – ‘Reseasrch check spelling, is it ‘research’?´
2.4. The proposed educational approach (lines 202-244)
Table 1 (line 199) and Table 2 (line 208) are too large, making the text that is included barely perceptible. Formatting with a different font size and setting is recommended.
3.2. Educational Experiment (lines 269-330)
Two different modules were included and applied to two different master's programs. It would be interesting to justify the reason for the application of two different modules since the aim of this study is to understand RQ:What pedagogical approaches and instructional methods need to be used to promote CT to computer science students?"
It remains to mention in which academic year the educational experiment took place and its duration. The ECTS are indicated but the contact hours and other working hours are not mentioned.
3.3. Instruments (lines 331-346)
To measure the impact of applying that Educational Experiment, two questionnaires were applied: qualitative (post-test) and quantitative (pre and post test)
It would be important to indicate the reference used for the elaboration of the qualitative data collection questionnaire similar to what was done for the quantitative questionnaires.
3.4. Data Collection and Analysis (lines 347-367)
It is mentioned that for data collection two experiments were carried out (lines 348-349). It is necessary to describe how those two experiments took place (the application of a module corresponds to an experiment?) and how the data were collected (pre and post test). How many students participated in these two experiments? Was the online questionnaire applied before and after the face-to-face experiment? How was the completion of these questionnaires monitored to ensure responses are reliable? Why so few students answered the quantitative data collection questionnaire? Were the qualitative and quantitative questionnaires not applied after the application of the experiments? The data collection process should be more explicit.
The two modules applied are quite different, so it is not understood why the qualitative data are categorized together to answer the research question.
4.Results (lines 368-469)
Enumeration in 4.1. is wrong and the percentages described in C1(91%; 55%), C2(all of them) and C3 (all of them) do not correspond to those in figure 1., so they must be checked. Line 435 replace “same number” by 55%
4.2. Students’ Self-assessment of Critical Thinking Skills and Dispositions (lines 447- 469)
Table 6 corresponds to the analysis of 20 data in pre test and 11 in post test but only 8 participated in both moments. What are the results for these students that can infer the impact of each experiment?
Line 459 – Only table 6
Table 7 uses data from the 5 students who participated in the quantitative and qualitative research, not indicating which master's degree they attended and therefore which educational experiment was applied.
5.Discussion
All results do not analyze the impact of each pedagogical approaches and instructional methods to conclude what is the best on. Why did the authors use different experiments if the data processing did not reflect this separation?
Author Response

(The authors gave the same response as above.)

Reviewer 3 Report
I found this to be an interesting and well-written paper on the important topic of learning critical thinking skills in the context of computational literacy instruction. A notable characteristic of the work is the use of mixed methods and the assessment both of learning and students' satisfaction, engagement, etc.
The most serious concern is both the small sample size and the fact that so many people were not assessed at the final testing. This raises the possibility that those students who were learning and/or enjoyed the program were the ones who stayed in, whereas those who were not doing as well dropped out. Thus we can't be sure whether the observed changes are due specifically to the program or the dropout rate. A related concern is the lack of a control of comparison group.
Author Response

(The authors gave the same response as above.)

Round 2
Reviewer 2 Report
The manuscript was significantly changed in response to identified needs for improvement in the first version
Author Response
Dear Reviewer,
thank you for you comments and the time you have invested in our manuscript.
Best regards